# Molecular Characterization of Porcine Epidemic Diarrhea Virus from Field Samples in South Korea

**DOI:** 10.3390/v15122428

**Published:** 2023-12-14

**Authors:** Bac Tran Le, Hansani Chathurika Gallage, Min-Hui Kim, Jung-Eun Park

**Affiliations:** College of Veterinary Medicine, Chungnam National University, Daejeon 34134, Republic of Korea; bac.le.dvm@gmail.com (B.T.L.); gallagehansani@gmail.com (H.C.G.); angelminhi10042@gmail.com (M.-H.K.)

**Keywords:** porcine epidemic diarrhea virus, spike, molecular characterization, phylogenetic analysis

## Abstract

Porcine epidemic diarrhea virus (PEDV) is a highly contagious enteric pathogen of swine. PEDV has been a major problem in the pig industry since its first identification in 1992. The aim of this study was to investigate the diversity, molecular characteristics, and phylogenetic relationships of PEDVs in field samples from Korea. Six PEDVs were identified from the field samples, and the full spike (S) glycoprotein gene sequences were analyzed. A phylogenetic analysis of the S gene sequences from the six isolates revealed that they were clustered into the G2b subgroup with genetic distance. The genetic identity of the nucleotide sequences and deduced amino acid sequences of the S genes of those isolates was 97.9–100% and 97.4–100%, respectively. A BLAST search for new PEDVs revealed an identity greater than 99.5% compared to the highest similarity of two different Korean strains. The CO-26K equivalent (COE) epitope had a 521H→Y/Q amino acid substitution compared to the subgroup G2b reference strain (KNU-1305). The CNU-22S11 had 28 amino acid substitutions compared to the KNU-1305 strain, which included two newly identified amino acid substitutions: 562S→F and 763P→L in the COE and SS6 epitopes, respectively. Furthermore, the addition and loss of N-linked glycosylation were observed in the CNU-22S11. The results suggest that various strains of PEDV are prevalent and undergoing evolution at swine farms in South Korea and can affect receptor specificity, virus pathogenicity, and host immune system evasion. Overall, this study provides an increased understanding of the prevalence and control of PEDV in South Korea.

## 1. Introduction

Porcine epidemic diarrhea (PED) is an acute and highly contagious enteric disease in swine caused by PED virus (PEDV). PED first appeared in England in 1971, manifesting as watery diarrhea and dehydration, resulting in mortality of up to 50% [1,2]. The causative agent of PED was identified in Belgium and the United Kingdom in 1978 [3], and since then, PEDV has been reported in many countries, especially in Europe and Asia [4,5]. In 2010, high-virulence PEDV variant strains emerged in China, resulting in high mortality and morbidity of up to 90–95% among infected sucking piglets [6]. In spring of 2013, a PEDV variant was confirmed in the US that was genetically close to the highly pathogenic Chinese strains and led to the massive deaths of newborn piglets within a year of the outbreak [7]. The strain rapidly spread to American, European, and Asian countries, initiating a second PED epidemiologic wave worldwide [8,9]. Thus, PEDVs were classified into two groups: classical G1, the strains of which have circulated since the 1970s, and highly pathogenic G2 variants, which emerged after the 2010s. Furthermore, PEDVs were divided into five subgroups. Subgroup G1b was identified after the attenuated vaccine for subgroup G1a (classical strains) was approved. In 2010, a highly pathogenic G2 PEDV with a genetic signature, spike insertions–deletions (S INDEL), appeared in China, and subgroup G2a PEDV showed high morbidity and mortality in piglets [6]. Group G2 continuously underwent evolution, and a new variant, subgroup G2b, was identified and spread around the world [10,11,12,13,14]. Subgroup G2c was generated as a result of the recombination of subgroups G1a and G2b [15].

PEDV, a member of the *alphacoronavirus* genus in the family *Coronaviridae*, is an enveloped virus, and the viral genome contains approximately 28 kb of single-stranded positive-sense RNA and encodes four structural proteins [16]. Similar to other coronaviruses, the spike (S) glycoprotein of PEDV plays an important role in the interaction with the host cell receptor during infection. The S protein contains the antigenic determinant regions that are targets of neutralizing antibodies in natural hosts [17]. Therefore, mutations occurring in the S gene can result in amino acid changes that affect pathogenicity, transmissibility, antigenic properties, and neutralizing antibody reactivity [18]. The molecular characterization of the S gene is beneficial for assessing the immune response to genetic variation.

In South Korea, PED was first reported in 1992 [19] and has since prevailed in many provinces, becoming one of the most important viral diarrhea diseases and resulting in heavy economic losses in the swine industry. To date, all five subgroups (1a, 1b, 2a, 2b, and 2c) have been reported in South Korea [20]. Vaccines derived from classical strains in the G1 group are highly effective against classical PEDV strains but have shown less protection against highly pathogenic strains in the G2 group [18]. Despite the use of vaccination, PEDV occurs in swine-raising farms in South Korea, bringing heavy losses due to continual PEDV infection. By analyzing the genetic information of prevalent PEDVs, we can prevent further outbreaks of PEDV-induced diarrhea more effectively and use the correct PEDV vaccine strains.

The present study was conducted to investigate the diversity among Korean PEDV isolates. We found more prevalent PEDVs in South Korea by sequencing analysis and determined the phylogenetic relationships of the S protein genes with PEDV isolates and reference strains.

## 2. Materials and Methods

### 2.1. Sample Collection

During 2022~2023, 50 porcine swabs and 28 fecal samples were collected from piglets exhibiting signs of watery diarrhea and dehydration. The sows of those piglets had been vaccinated with commercial vaccines used in South Korea. These samples were screened for four viruses that can cause porcine diarrhea: porcine epidemic diarrhea virus (PEDV), transmissible gastroenteritis virus (TGEV), porcine deltacoronavirus (PDCoV), and porcine rotavirus by singleplex RT–PCR [21]. Briefly, the samples were diluted in phosphate-buffered saline to 10% suspensions. The suspensions were vortexed and clarified by centrifugation for 30 min at 2000× *g*. The supernatants were collected for RNA preparation using a QIAamp Viral RNA Mini Kit (Qiagen, Valencia, CA, USA) in accordance with the manufacturer’s instructions. The RNA was then subjected to PEDV screening or amplification of the S protein-encoding gene for molecular analysis.

### 2.2. RT–PCR and DNA Cloning

cDNA was synthesized from viral RNA (1 µg) using a PrimeScript™ First-Strand cDNA Synthesis Kit following the manufacturer’s instructions (Takara, Japan), and PCR was performed targeting the 2 segments covering the full open reading frame (ORF) coding regions of the S gene using Premix Taq™ (Takara, Japan). The primers were named as follows: PEDV-S-Full-F (5′-GCT AGT GCG TAA TAA TGA CGC CA-3′); PEDV-S1-R (5′-ACA GAG CCT GTG TTG GTG TA-3′); PEDV-S2-F (5′-TAC TAG GGA GTT GCC TGG TT-3′), and PEDV-S-Full-R (5′-AGG TCC ACG TGC AGT GAT GT-3′). These primers were used for amplification of the S1 and S2 segments of new PEDV isolates. The PCR mixture contained 25 μL of premix Taq 2× buffer, 1 μL of each primer (10 pmol), 3 μL of cDNA template, and a final water reaction volume of 50 μL. The PCR thermocycling procedure was performed at 98 °C for 20 s, 56 °C for 30 s, and 72 °C for 120 s for 35 cycles. The S1 and S2 segments overlapped for 72 bp in the full S assembly. The amplicons were separated by gel electrophoresis. The bands were then excised from the gel and purified using the QIAquick Gel Extraction Kit (Qiagen, CA, USA). The purified DNA was cloned and inserted into the pGEM-T vector (Promega, Madison, WI, USA) and transformed into DH5-alpha competent cells (Enzynomics, Daejeon, Republic of Korea) according to the manufacturer’s instructions.

### 2.3. Sequence Analysis

Three plasmids of each successful clone were subjected to sequencing using an ABI3730XL DNA analyzer (Cosmo Genetech, Seoul, Republic of Korea). The full S gene was assembled using the Lasergene sequence analysis software package (DNA Star version 4.0, Madison, WI, USA). Noncoding regions containing primer sequences were trimmed. A BLAST search of the GenBank database was used to determine the strains most closely related to the new viruses. The new isolates and PEDV reference strains used for nucleotide sequence analysis and amino acid deduction sequence analysis in this study are described in Table 1. N-linked glycosylation was predicted as previously stated [22].

### 2.4. Multiple Alignment and Phylogenetic Analyses

The inputs of entire ORF sequences of the S gene included both the 6 new isolates and 57 reference strain sequences from the GenBank database, an open-access resource. The sequences were aligned by multiple alignment using the ClustalW algorithm in BioEdit software (Version 7.0). Phylogenetic analysis of S gene sequences was performed using Molecular Evolutionary Genetics Analysis Version 11.0 software (MEGA 11.0). The evolutionary distances were computed using the neighbor-joining method. Statistical values greater than 60%, a measure of reliability from bootstrap (*n* = 1000) iterations, are shown.

## 3. Results

### 3.1. PEDV Detection and S Gene Sequencing

A total of 6 of the 78 field samples (7.7%) were determined to contain PEDV sequences. Information relating to the six positive samples is provided in Appendix A. None of the fecal samples or fecal swabs showed any evidence of sequences from TGEV, PDCoV, or porcine rotavirus. The complete S gene was amplified using two primer pairs: PEDV-S-Full-F, PEDV-S1-R and PEDV-S2-F, PEDV-S-Full-R, and the genes were sequenced using Sanger sequencing. The S genes of CNU-22P1, CNU-22P2, CNU-22P4, CNU-22S11, CNU-22S16, and CNU-22S17 were discovered to be 4161 nucleotides in length and submitted to GenBank under accession numbers OR529200-OR529205.

### 3.2. Phylogenetic Analysis of the S Genes

The six new PEDV isolates and 57 PEDV reference strains from GenBank were subjected to phylogenetic analysis to understand the phylogenetic relationships. The reference strains represented five subgroups belonging to two classic groups, G1 and the variant group G2 (Table 1). The CV777 and DR13 wild-type strains are in the G1a subgroup. The subgroups G1b, G2a, G2b, and G2c represent the attenuated vaccine strains (CV777 and DR13), AJ1102, KNU-1305, and OH851, respectively. Based on the full S genetic tree, all six isolates identified in this study belong to the G2b subgroup and are closely related to strains previously isolated in South Korea during 2021~2022 (Figure 1A). The phylogenetic trees of the S1 (Figure 1B) and S2 (Figure 1C) subunits showed similar patterns to that of the full S gene.

### 3.3. S Gene Sequence Comparative Analysis

The S protein genes of the six new PEDVs consists of 4161 nucleotides encoding a 1386 amino-acid protein. The genetic identities of the nucleotide and amino acid sequences of the S gene reflected 97.9~100% and 97.4~100% similarity between isolates, respectively. The S protein-encoding gene is divided into 2 subunits, S1 and S2. The S1 subunit consists of 2187 nucleotides and encodes a protein 729 amino acid residues in length. The S1 subunit shared nucleotide and amino acid identities of 97.5~100% and 96.9~100% among these new PEDVs, respectively. The S2 subunit consists of 1974 nucleotides and codes for 657 amino acid residues, starting at residue 729 at the C-terminus. The S2 subunit shared nucleotide and amino acid identities of 98.2~100% and 97.8~100% among the new PEDVs, respectively (Table 2). The S protein-encoding gene of the CNU-22S11 had the highest diversity, with less than 98% similarity to other isolates and 30 amino acid substitutions compared to the reference strain. A BLAST search was conducted to identify the strains with the highest similarity with the new PEDVs. Two isolated Korean PEDV strains showed similarity greater than 99.5%, which are underlined in Table 2. Based on the S1 subunits, the new PEDVs shared nucleotide and amino acid identities of 97.8~98.9% and 97.1~98.7% to subgroup G2b corresponding strain (KNU-1305), which is underlined in the gray box in Table 2. Compared to commonly used vaccine strains in Korea (CV777, DR13), the isolates shared less than 94% nucleotide and amino acid identities in the full S gene.

### 3.4. Feature Amino acid Deduction Analysis

There were four defined epitopes in isolates at amino acid positions 499CO-26K equivalent638 (COE), 748YSNIGVCK755 (SS2), 763(L/P)SQSGQVKI771 (SS6), and 1368GPRLQPY1374 (2C10) [23,24,25]. The COE epitope had 521Y/H/Q and 562F/S amino acid substitutions among the different isolates (Figure 2). Following the vaccine CV777 strain (AF353511) numbering, the isolates presented two insertions (at amino acids 57NQGV and 135N), one deletion (at amino acid 158DI159), 43 amino acid substitutions in the N-terminal domain, and 9 amino acid substitutions in the COE epitope in all new isolates, 517A→S, 521L→Q/Y/H, 523S→G, 527V→I, 549T→S, 562S→F, 594G→S, 605A→E, 612L→F, and 635I→V compared to the CV777 strain (Figure 2). Four or five amino acid substitutions were found in five out of six isolates, with the exception of the CNU-22S11 compared to the KNU-1305 strain (Figure 2). For the CNU-22S11, there were 28 amino acid substitutions, including 3 amino acid substitutions, 562S→F, 763P→L, and 1330I→K, in the COE and SS6 epitope and transmembrane regions, respectively (Figure 2).

The N-linked glycosylation sites were predicted based on the consensus N–X–S/T glycosylation motif (X can be any amino acid except proline). As a result, the six PEDVs had 29 predicted N-glycosylation sites, similar to the CV777 and KNU-1305 strains. The isolates showed a more similar pattern to KNU-1305 (Table 3). However, N-linked glycosylation prediction revealed the addition of one glycosylation site at residue 722 and the deduction of one glycosylation site at residue 1232 of CNU-22S11 compared to KNU-1305 (Table 3).

## 4. Discussion

To date, all five PEDV subgroups belonging to the G1 and G2 groups are present in South Korea. The emergence of the PEDV G1 lineage was first reported in 1992 [19]. Since then, PEDV has become endemic in the domestic swine population, as detected by serological tests among weaned and growing–finishing pigs on more than 90% of farms in 2007 [26]. The highly pathogenic G2 group appeared in South Korea in the 2010s, and the G2b subgroup entered South Korea soon after the outbreak in the US and swept through almost 50% of the pig farms nationwide (Jeju Island was not an exception) [11,27]. The novel recombinant G2c subgroup was the result of the recombination of a minor parent G1 and a major parent G2b, which was first reported in South Korea in 2014 and has been reported previously in China and the US [6,14,28]. G2c has minor impacts on the pig industry [28]. However, the coexistence of the G2c and dominant G2b groups leads to the generation of inner-subgroup G2c variants [29]. The six PEDVs identified in this report belonged to the G2b subgroup. We also analyzed the similarity of our isolates to recently published data from South Korea [30] and neighboring countries, such as China [31], Taiwan [32], and Vietnam [33]. The isolates shared over 99.5% similarity among South Korean isolates. However, the isolates shared less than 98.5% similarity, especially CNU-22S11, which shared less than 97.5% similarity with strains in neighboring countries. Along with previous research, our data added value, showing that the G2b subgroups, with further distinct evolution, have become dominant strains in pig farms in South Korea in recent years [30].

In addition, the new CNU-22S11 in this study showed less genetic similarity to other isolates and subgroup reference strains (KNU-1305). Several amino acid substitutions were identified in the N-terminal domain of the S1 protein subunit. In particular, a number of new amino acid substitutions in the COE and SS6 epitopes were observed. Mutation may affect the spatial conformations of the neutralizing epitopes of the S protein and the ability of the host to produce neutralizing antibodies [18]. Furthermore, the addition and loss of glycosylation sites were observed in CNU-22S11. The addition of one glycosylation site at residue 722 is common in PEDV isolates in South Korea. Remarkably, the loss of one glycosylation site at residue 1232 was found only in a PEDV strain (GenBank Acc. No. KJ857479) in 2013. There is a novel finding in both the addition and loss of glycosylation sites in the CNU-22S11. Viral glycoproteins are modified by the host cell’s N-linked glycosylation pathways; in particular, the S glycoprotein plays an important role in the generation of host innate responses toward the virus [34]. Our findings suggest that mutations can affect receptor specificity, virus pathogenicity, and host immune system evasion.

In conclusion, our data reveal that the circulation of PEDV in South Korea is continuing to evolve independently, and its genetic diversity is increasing. This study further expands our understanding of the molecular epidemiology of PEDV in South Korea. Our findings highlight the critical need to monitor the circulation of PEDV and develop novel vaccines against recently emerged PEDV variants.

## Figures and Tables

**Figure 1 viruses-15-02428-f001:**
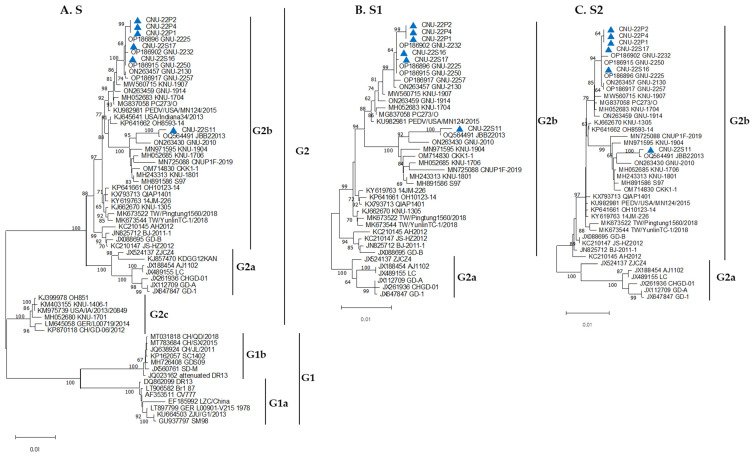
Phylogenetic relationships based on the full S (**A**); S1 (**B**); and S2 (**C**) glycoprotein genes of PEDV isolates and PEDV reference strains. Phylogenetic trees were constructed using Molecular Evolutionary Genetics Analysis Version (MEGA) 11.0. The evolutionary distances were computed using the neighbor-joining method, and the statistical values greater than 60% of the measure of reliability from the bootstrap (*n* = 1000) iterations are shown. The analyzed isolates in this study are labeled with blue triangles.

**Figure 2 viruses-15-02428-f002:**
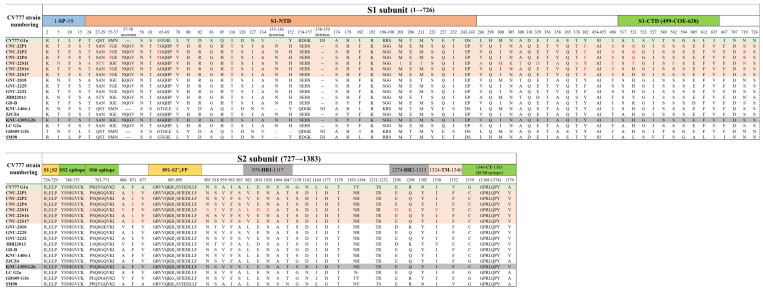
Schematic representation of specific domains of the PEDV S protein and insertion–deletion–substitution analysis. The S protein consists of two subdomains, the S1 and S2 subunits. The S1 subunit contains a signal peptide (SP) and two receptor-binding domains: the N-terminal domain (S1-NTD) and the C-terminal domain (S1-CTD) covering the CO-26K equivalent (COE) region. The S2 subunit contains three neutralization epitopes (SS2, SS6, and 2C10), two heptad repeat domains (HR1 and HR2), a transmembrane domain (TM), and a cytoplasmic domain (CT). The orange-shadowed boxes are the isolates identified in this study, and the green-shadowed and gray-shadowed boxes are subgroup reference strains (CV777 and KNU-1305). The arrows indicate proteolytic cleavage sites. The dashes (-) indicate regions of insertion or deletion relative to CV777. The red letters indicate amino acid substitutions compared to the KNU-1305 strain.

**Table 1 viruses-15-02428-t001:** Representative strains and the distribution and time of appearance of each PEDV genotype.

Genotype	Variation	Representative Strains	Distribution	First Appearance	Recorded in South Korea
G1a	Classical strains	CV777; DR13; LZC; SM98; GER L00901-V215	Asia; Europe	1971; England	1992
G1b	Related to G1a S INDEL attenuated vaccine strains	attenuated CV777; attenuated DR13; SD-M; SC1402	Asia	2000s; Asia	2001
G2a	S non-INDEL variant strains	AJ1102; LC; CHGD-01; GD-1; GD-A; ZJCZ4; KDGG12KAN	Asia	2010; China	2012
G2b	S non-INDEL variant strains	AH2012; BJ-2011-1; GD-B; KNU-1305; TW/Pingtung1560; Indiana34; MEX/104/2013	Asia; North America	2011; China	2013
G2c	S INDEL strains recombined between G1a and G2b	CH/GD-06/2012; KNU-1406-1; OH851; GER/L00719/2014	Asia; North America; Europe	2012; China	2014

**Table 2 viruses-15-02428-t002:** Homology of the nucleotide and deduced amino acid sequences of the S1, S2, and full S protein genes of PEDV isolates and PEDV reference strains. Nucleotide identity (%) in bold; deduced amino acid identity (%) in italics; highest identities of the PEDV isolates to the BLAST-searched strains and the PEDV group corresponding to the reference strains are underlined; the gray-shadowed box corresponds to the reference strains for each group.

Strain	CNU-22P1						CNU-22P2						CNU-22P4						CNU-22S11						CNU-22S16						CNU-22S17					
	S1		S2		Full S		S1		S2		Full S		S1		S2		Full S		S1		S2		Full S		S1		S2		Full S		S1		S2		Full S	
**CNU-22P1**	ID		ID		ID																															
**CNU-22P2**	**100**	*100*	**100**	*100*	**100**	*100*	ID		ID		ID																									
**CNU-22P4**	**100**	*100*	**100**	*100*	**100**	*100*	**100**	*100*	**100**	*100*	**100**	*100*	ID		ID		ID																			
**CNU-22S11**	**97.5**	*96.9*	**98.2**	*97.8*	**97.9**	*97.4*	**97.5**	*96.9*	**98.2**	*97.8*	**97.9**	*97.4*	**97.5**	*96.9*	**98.2**	*97.8*	**97.9**	*97.4*	ID		ID		ID													
**CNU-22S16**	**99.8**	*99.7*	**99.9**	*100*	**99.8**	*99.8*	**99.8**	*99.7*	**99.9**	*100*	**99.8**	*99.8*	**99.8**	*99.7*	**99.9**	*100*	**99.8**	*99.8*	**97.7**	*97.1*	**98.3**	*97.8*	**98**	*97.4*	ID		ID		ID							
**CNU-22S17**	**99.7**	*99.7*	**99.9**	*100*	**99.8**	*99.8*	**99.7**	*99.7*	**99.9**	*100*	**99.8**	*99.8*	**99.7**	*99.7*	**99.9**	*100*	**99.8**	*99.8*	**97.7**	*97.1*	**98.3**	*97.8*	**98**	*97.4*	**99.9**	*100*	**100**	*100*	**99.9**	*100*	ID		ID		ID	
**GNU-2010**	**97.9**	*97.3*	**98.5**	*97.7*	**98.2**	*97.5*	**97.9**	*97.3*	**98.5**	*97.7*	**98.2**	*97.5*	**97.9**	*97.3*	**98.5**	*97.7*	**98.2**	*97.5*	**98.7**	*97.8*	**99**	*98.3*	**98.8**	*98*	**98.1**	*97.6*	**98.5**	*97.7*	**98.3**	*97.6*	**98**	*97.6*	**98.5**	*97.7*	**98.3**	*97.6*
**GNU-2225**	** 99.8 **	** * 99.7 * **	** 99.9 **	** * 100 * **	** 99.8 **	** * 99.8 * **	** 99.8 **	** * 99.7 * **	** 99.9 **	** * 100 * **	** 99.8 **	** * 99.8 * **	** 99.8 **	** * 99.7 * **	** 99.9 **	** * 100 * **	** 99.8 **	** * 99.8 * **	**97.7**	*97.1*	**98.3**	*97.8*	**98**	*97.4*	** 100 **	** * 100 * **	** 100 **	** * 100 * **	** 100 **	** * 100 * **	** 99.9 **	** * 100 * **	** 100 **	** * 100 * **	** 99.9 **	** * 100 * **
**GNU-2232**	**99.8**	*99.7*	**99.7**	*99.8*	**99.8**	*99.7*	**99.8**	*99.7*	**99.7**	*99.8*	**99.8**	*99.7*	**99.8**	*99.7*	**99.7**	*99.8*	**99.8**	*99.7*	**97.7**	*97.1*	**98.1**	*97.7*	**97.9**	*97.4*	**100**	*100*	**99.8**	*99.8*	**99.9**	*99.9*	**99.9**	*100*	**99.8**	*99.8*	**99.9**	*99.9*
**JBB22013**	**97.8**	*97.5*	**98.3**	*98.1*	**98**	*97.8*	**97.8**	*97.5*	**98.3**	*98.1*	**98**	*97.8*	**97.8**	*97.5*	**98.3**	*98.1*	**98**	*97.8*	** 99.7 **	** * 99.4 * **	** 99.8 **	** * 99.6 * **	** 99.8 **	** * 99.5 * **	**97.9**	*97.6*	**98.3**	*98.1*	**98.1**	*97.9*	**97.9**	*97.6*	**98.3**	*98.1*	**98.1**	*97.9*
**GD-B**	**98**	*98.2*	**99.2**	*99.5*	**98.6**	*98.8*	**98**	*98.2*	**99.2**	*99.5*	**98.6**	*98.8*	**98**	*98.2*	**99.2**	*99.5*	**98.6**	*98.8*	**97**	*96.5*	**98.3**	*98*	**97.6**	*97.2*	**98.2**	*98.3*	**99.3**	*99.5*	**98.7**	*98.9*	**98.1**	*98.3*	**99.3**	*99.5*	**98.7**	*98.9*
**ZJCZ4**	**98**	*98*	**97.9**	*98.7*	**98**	*98.4*	**98**	*98*	**97.9**	*98.7*	**98**	*98.4*	**98**	*98*	**97.9**	*98.7*	**98**	*98.4*	**97.1**	*96.5*	**97.2**	*97.5*	**97.1**	*97*	**98.2**	*98.3*	**98**	*98.7*	**98.1**	*98.5*	**98.1**	*98.3*	**98**	*98.7*	**98.1**	*98.5*
**KNU-1406-1**	**92.6**	*91.9*	**99.5**	*99.6*	**95.8**	*95.6*	**92.6**	*91.9*	**99.5**	*99.6*	**95.8**	*95.6*	**92.6**	*91.9*	**99.5**	*99.6*	**95.8**	*95.6*	**92.2**	*90.9*	**98.6**	*98.1*	**95.2**	*94.3*	**92.7**	*92*	**99.5**	*99.6*	**95.9**	*95.6*	**92.7**	*92*	**99.5**	*99.6*	**95.9**	*95.6*
**SM98**	**89.9**	*88.9*	**94.5**	*94.6*	**92.1**	*91.6*	**89.9**	*88.9*	**94.5**	*94.6*	**92.1**	*91.6*	**89.9**	*88.9*	**94.5**	*94.6*	**92.1**	*91.6*	**89.3**	*88.1*	**93.7**	*93.6*	**91.4**	*90.7*	**90.1**	*88.9*	**94.5**	*94.6*	**92.2**	*91.6*	**90**	*88.9*	**94.5**	*94.6*	**92.1**	*91.6*
**KNU-1305 G2b**	** 98.9 **	** * 98.7 * **	** 99.4 **	** * 99.3 * **	** 99.1 **	** * 99 * **	** 98.9 **	** * 98.7 * **	** 99.4 **	** * 99.3 * **	** 99.1 **	** * 99 * **	** 98.9 **	** * 98.7 * **	** 99.4 **	** * 99.3 * **	** 99.1 **	** * 99 * **	** 97.8 **	** * 97.1 * **	** 98.5 **	** * 97.8 * **	** 98.1 **	** * 97.4 * **	** 99 **	** * 99 * **	** 99.5 **	** * 99.3 * **	** 99.3 **	** * 99.2 * **	** 99 **	** * 99 * **	** 99.5 **	** * 99.3 * **	** 99.2 **	** * 99.2 * **
**LC G2a**	**97.7**	*97.9*	**97.1**	*98.1*	**97.4**	*98*	**97.7**	*97.9*	**97.1**	*98.1*	**97.4**	*98*	**97.7**	*97.9*	**97.1**	*98.1*	**97.4**	*98*	**96.8**	*96.4*	**96.3**	*96.9*	**96.6**	*96.6*	**97.9**	*98.2*	**97.2**	*98.1*	**97.5**	*98.1*	**97.8**	*98.2*	**97.2**	*98.1*	**97.5**	*98.1*
**GDS09 G1b**	**90.1**	*89.4*	**96.1**	*95.8*	**92.9**	*92.5*	**90.1**	*89.4*	**96.1**	*95.8*	**92.9**	*92.5*	**90.1**	*89.4*	**96.1**	*95.8*	**92.9**	*92.5*	**89.4**	*88.5*	**95.4**	*95.1*	**92.2**	*91.6*	**90.1**	*89.6*	**96.2**	*95.8*	**93**	*92.5*	**90.1**	*89.6*	**96.2**	*95.8*	**93**	*92.5*
**CV777 G1a**	**90.6**	*90*	**96**	*96.6*	**93.2**	*93.1*	**90.6**	*90*	**96**	*96.6*	**93.2**	*93.1*	**90.6**	*90*	**96**	*96.6*	**93.2**	*93.1*	**90**	*89.1*	**95.2**	*95.5*	**92.5**	*92.2*	**90.7**	*90*	**96**	*96.6*	**93.3**	*93.1*	**90.7**	*90*	**96**	*96.6*	**93.2**	*93.1*

**Table 3 viruses-15-02428-t003:** Predicted N-linked glycosylation sites based on the consensus N–X–S/T (X can be any amino acid except proline) glycosylation motif. The asterisks indicate stop codons.

CV777 numbering	57			127	213		230	261		297		321		341		348		378		422		511		553		664		685		719		723
CV777 G1a	NSS			NKT	NVT		NCT	NDS		NHT		NDT		NLS		NSS		NST		NFT		NIT		NVT		NSS		NVT		NST		NNT
CNU-22P1		NST	NAT			NVT			NDS		NQT		NDT		NLS		NSS		NST		NFT		NIT		NVT		NSS		NVT			
CNU-22P2		NST	NAT			NVT			NDS		NQT		NDT		NLS		NSS		NST		NFT		NIT		NVT		NSS		NVT			
CNU-22P4		NST	NAT			NVT			NDS		NQT		NDT		NLS		NSS		NST		NFT		NIT		NVT		NSS		NVT			
CNU-22S11		NST	NAT			NVT			NDS		NQT		NDT		NLS		NSS		NST		NFT		NIT		NVT		NSS		NVT		NST	
CNU-22S16		NST	NAT			NVT			NDS		NQT		NDT		NLS		NSS		NST		NFT		NIT		NVT		NSS		NVT			
CNU-22S17		NST	NAT			NVT			NDS		NQT		NDT		NLS		NSS		NST		NFT		NIT		NVT		NSS		NVT			
KNU-1305 G2b		NST	NAT			NVT			NDS		NQT		NDT		NFS		NSS		NST		NFT		NIT		NVT		NSS		NVT			
KNU-1305 numbering		62	118			216			264		300		324		344		351		381		425		514		556		667		688		722	
CV777 numbering	740		778		784		870		1006			1229		1246		1258		1270		1275		1292		1305		1384						
CV777 G1a	NCT		NIS		NFS		NFT		NIT			NLT		NKT		NRT		NAT		NLT		NTT		NNT		*						
CNU-22P1		NCT		NIS		NFS		NLT		NIT	NHT		NLT		NKT		NRT		NAT		NLT		NTT		NNT	*						
CNU-22P2		NCT		NIS		NFS		NLT		NIT	NHT		NLT		NKT		NRT		NAT		NLT		NTT		NNT	*						
CNU-22P4		NCT		NIS		NFS		NLT		NIT	NHT		NLT		NKT		NRT		NAT		NLT		NTT		NNT	*						
CNU-22S11		NCT		NIS		NFS		NFT		NIT	NHT				NKT		NRT		NAT		NLT		NTT		NNT	*						
CNU-22S16		NCT		NIS		NFS		NLT		NIT	NHT		NLT		NKT		NRT		NAT		NLT		NTT		NNT	*						
CNU-22S17		NCT		NIS		NFS		NLT		NIT	NHT		NLT		NKT		NRT		NAT		NLT		NTT		NNT	*						
KNU-1305 G2b		NCT		NIS		NFS		NFT		NIT	NHT		NLT		NKT		NRT		NAT		NLT		NTT		NNT	*						
KNU-1305 numbering		743		781		787		873		1009	1196		1232		1249		1261		1273		1278		1295		1308	1387						

## Data Availability

The data presented in this study are available on request from the corresponding author.

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
