# Peer review of "Molecular Characterization of Porcine Epidemic Diarrhea Virus from Field Samples in South Korea"

_viruses, 2023, doi:10.3390/v15122428_

Round 1
Reviewer 1 Report
Comments and Suggestions for Authors
The work concerns the molecular analysis of 6 PEDV isolates obtained from piglets from South Korea. Work brings nothing new. Single mutations were detected, but for coronaviruses that constantly mutate this is a normal feature. The paper can be considered as short communication.
Specific comments:
1. Line 11- in line 33 authors claim that first identification was in 1971 in England.
2. Lines34-35. In the previous sentence, the authors stated that the first identification of the virus in the England was in 1971.
3. Please check the division of PEDV into groups. It seems to me that group G1 includes strains S-INDEL and group G2 includes non-S INDEL strains.
4. details of amplification conditions are missing in section 2.2.
5. line 120. In the methodology, it is not mentioned that the samples were tested for TGEV, PDCoV and porcine rotavirus.
6. “The S1 subunit shared nu and amino acid identity of 97.5%-100% and 96.9-100% among of these new strains, respectively” nucleotide instead nu
7. Please check the nucleotide and amino acid identity for S protein , S1 and S2 because nucleotide identity is higher than amino acid identity. This is quite strange.
8. Materials and methods lacks a description of how N-linked glycosylation sites were predicted.
9. On what basis were strains selected whose sequences were compared with those of the strains tested in section 3.4?
10. For me, table 3 is incomprehensible.
11. Discussion is too short.
Reviewer 2 Report
Comments and Suggestions for Authors
In this manuscript, Le, B.T. et al. described the variant strains of PEDV in South Korea. The authors collected the different PEDV samples from the field and sequenced them for their genetic differentiation. The classification of the newly identified PEDV strains contains many mutations on the spike COE and S1 region, which may allow the virus to escape from the vaccine. Overall, the conceptual idea is interesting and points to an important issue for the undergoing evolution of PEDV infectious disease in South Korea. However, some experimental and conceptual issues need to be clarified:
1. The results showed that CNU-22S11 has significant genetic differences and that the loss of N-linked glycosylation might affect receptor specificity, virus pathogenicity, and host immune system evasion. This inference can be strengthened by adding the serologic test (Ex: ELISA, or Western blot) to test whether the CNU-22S11 strain can escape from the original vaccine (CV-777) and the host immune system.
2. In this manuscript, Six PEDV strains were identified from the field samples based on spike (S) glycoprotein gene sequence. Phylogenetic analysis of the S gene sequences revealed that they are clustered into the G2b subgroup. Although the authors cited previous references about other G2b strains, the authors should discuss the relationship of the newly identified South Korean strains to those identified from other countries, especially from surrounding Asian countries.
3. The authors should also discuss the possible differences in sequences or protein structure making the vaccines against classical G1 strain not valid against the highly pathogenic G2 variant that emerged after the 2010s.
Comments on the Quality of English LanguageThe English need moderate revision. Here are some examples:
Line 36, PEDV variant strains emerged in China resulted-->
PEDV variant strains emerged in China resulting
Line 37, In spring 2013, a variant PEDV was confirmed -->
In the spring of 2013, a variant of PEDV was confirmed
Line 39, to the massive deaths of newborn piglets in a year of the outbreak. -->
to massive deaths of newborn piglets within a year of the outbreak.
Reviewer 3 Report
Comments and Suggestions for Authors
The Porcine Epidemic Diarrhea Virus (PEDV) is a highly contagious swine enteric pathogen that poses significant challenges to the pig industry. The authors of this study attempted to investigate the diversity, molecular characteristics, and phylogenetic relationships of recent PEDV isolates. They isolated six PEDV isolates from 78 fecal swab or feces samples collected in South Korea from 2022 to 2023. The spike (S) glycoprotein gene sequence analysis and phylogenetic analysis placed these isolates in the G2b subgroup. Six isolates' S gene nucleotide and amino acid sequences showed high genetic identity (97.9-100% and 97.4%-100%, respectively) and were closely related to existing Korean strains. In addition, the CNU-22S11 isolate contains 28 amino acid substitutions, including two novel ones. The experiment was carefully carried out, and the results add to our understanding of the prevalence and evolution of various PEDV strains in South Korean swine farms. The number of isolates, however, is limited. It would be interesting to compare the findings to those of a recent paper (Park GN, et al. Spike Gene Analysis and Prevalence of Porcine Epidemic Diarrhea Virus from Pigs in South Korea:2013-2022. Viruses. 2023 Oct 28;15(11):2165). I also have some concerns that need to be addressed.
1. The collection time and region(s), particularly for isolates from 2023, should be more precisely described. Furthermore, the list of vaccines used on the farm should be described in the manuscript.
2. What is the DNA polymerase used in PCR?
3. The protocol used to detect TGEV, PDCoV, or porcine rotavirus should be described in the materials and methods section.
4. Section 3.3, line 8, "The S1 subunit shared nu and amino acid identity of", the meaning of “nu” needs to be clarified.
5. In Figure 2, the abbreviations "S1-NDT" and "S1-CDT" in the figure differ from the description in the legend. Please go over them thoroughly.
6. The term "new stains" used in the manuscript is imprecise; the S gene of CNU-22P1, P2, and P4 appears to be identical; "isolates" appears to be a more appropriate term in the manuscript.
Comments on the Quality of English LanguageI recommend English editing to improve clarity and readability.
